# Synthesis and Application of a New Polymer with Imprinted Ions for the Preconcentration of Uranium in Natural Water Samples and Determination by Digital Imaging

**DOI:** 10.3390/molecules28104065

**Published:** 2023-05-12

**Authors:** Caio S. A. Felix, Adriano V. B. Chagas, Rafael F. de Jesus, Willams T. Barbosa, Josiane D. V. Barbosa, Sergio L. C. Ferreira, Víctor Cerdà

**Affiliations:** 1Instituto de Química, Programa de Pós-Graduação em Química, Campus Ondina, Universidade Federal da Bahia, Salvador 40170-115, Brazil; caio_silva_@hotmail.com (C.S.A.F.); adrianovchagas@hotmail.com (A.V.B.C.); rafaeelfranco13@gmail.com (R.F.d.J.); slcf@ufba.br (S.L.C.F.); 2Centro Interdiciplinar de Energia e Ambiente—CIEnAm, Universidade Federal da Bahia, Salvador 40170-110, Brazil; 3SENAI CIMATEC University Center, Programa de Pós-Graduação em Gestão e Tecnologia Industrial, Salvador 41650-010, Brazil; willams.barbosa@fbter.org.br (W.T.B.); josianedantas@fieb.org.br (J.D.V.B.); 4Department of Chemistry, University of the Balearic Islands, 07122 Palma de Mallorca, Spain

**Keywords:** uranium, waters, digital imaging, imprinted ions

## Abstract

This work proposes the synthesis of a new polymer with imprinted ions (IIP) for the pre-concentration of uranium in natural waters using digital imaging as a detection technique. The polymer was synthesized using 2-(5-bromo-2-pyridylazo)-5-diethylaminophenol (Br-PADAP) for complex formation, ethylene glycol dimethacrylate (EGDMA) as a crosslinking reagent, methacrylic acid (AMA) as functional monomer, and 2,2′-azobisisobutyronitrile as a radical initiator. The IIP was characterized by Fourier transform infrared spectroscopy and scanning electron microscopy (FTIR). Uranium determination was performed using digital imaging (ID), and some experimental conditions (sample pH, eluent concentration, and sampling flow rate) were optimized using a two-level full factorial design and Doelhert response surface methodology. Thus, using the optimized conditions, the system allowed the determination of uranium with detection and quantification limits of 2.55 and 8.51 µg L^−1^, respectively, and a pre-concentration factor of 8.2. All parameters were determined using a 25 mL sample volume. The precision expressed as relative deviation (RSD%) was 3.5% for a solution with a concentration of 50 µg L^−1^. Given this, the proposed method was used for the determination of uranium in four samples of natural waters collected in the city of Caetité, Bahia, Brazil. The concentrations obtained ranged from 35 to 75.4 μg L^−1^. The accuracy was evaluated by the addition/recovery test, and the values found ranged between 91 and 109%.

## 1. Introduction

With industrialization in modern society, the problem of carbon dioxide emissions and global warming of the planet has increased [1,2]. As a result, other sources of clean energy have been used to minimize these harmful effects on the environment. An example of this is atomic energy [2,3].

Uranium, which is primarily used as fuel in nuclear power plants, is found in soil and water. It is known to cause serious toxicological effects to humans and its compounds are carcinogenic. It reaches the environment by leaching from natural deposits, releasing waste from nuclear power plants, and the combustion of coal and other fuels [4,5].

Currently, the separation and determination of uranium are gaining more and more importance due to its increasing applications in different fields, its use goes beyond the generation of nuclear energy, also being applied in the industry of anti-tank ammunition, nuclear weapons, catalysts, and pigments of dyes [6]. According to the World Nuclear Association (WNA), it is estimated that there are 6.14 million tons of uranium in mines around the world. From this amount, Brazil represents about 5% [5,7].

There are several reported techniques capable of detecting uranium in environmental samples, such as: fluorometry [8,9], X-ray fluorescence [10,11], potentiometry [12,13], inductively coupled plasma emission spectrometry (ICP OES) [14,15,16], inductively coupled plasma mass spectrometry (ICP-MS) [17,18], and spectrophotometry [5,19,20,21,22,23]. Among all these techniques, spectrophotometry has a lot of applications. In recent years, many conventional analytical methods based on computer vision related to the colorimetric analysis of digital images are becoming promising, allowing the development of fast, accurate, and low-cost analytical methodologies [24]. Some detectors such as scanners [25,26,27], digital cameras [28,29], smartphones [30,31,32], tablet computers, and webcams [33,34] can be used.

Given this infinity of cited techniques, interference from other ions and low sensitivity are the most common problems to be overcome by the proposed methodologies. For this, it is necessary to create methods that allow the pre-concentration of uranium ions. To date, there are no reports in the literature about methodologies that involve the preconcentration and determination of uranium by digital images. Ion-imprinted polymers (IIPs) have advantageous characteristics to overcome these disadvantages, such as high selectivity, greater stability in aggressive environments, such as high temperatures and pressures, and ease of synthesis [5,35,36].

This work proposes in an unprecedented way, so far, the application of a pre-concentration method using IIP-U as a new sorbent and the quantification using digital images of uranium in natural water samples from the city of Caetité (Bahia, Brazil).

## 2. Results and Discussion

### 2.1. Choice of RGB Channel

A study was carried out in order to determine the best RGB channel for the analysis using the digital image as a way of detecting the uranium concentrations of the proposed method. For this, an analytical curve was performed with concentrations varying between 0 and 2 mg L^−1^. From the ROI, the images were decomposed into RGB signals with the help of the ImageJ software, and using Equation (1), the RGB values were converted to absorbance values. The result of this test can be seen in Table 1.

With the absorbance values calculated from Equation (1), it was possible to plot the analytical curves shown in Figure 1. Observing them, it can be seen that the channel that has the best correlation coefficient is the R (red) channel (R^2^ = 0.9761) as well as greater sensitivity expressed by the angular coefficient (0.2056). Therefore, this channel was chosen to carry out the other experiments [25].

### 2.2. Arsenazo III Concentration Study

A univariate study was carried out in order to know the optimal concentration of arsenazo III to be used for the formation of the complex with U(VI). The result can be seen in Figure 2. From this study, it was observed that the optimal concentration of arsenazo III for the formation of the U(VI)–arsenazo III complex is 0.005% (*m*/*v*).

### 2.3. Synthesis and Characterization of the Ion-Imprinted Polymer (IIP)

The uranyl (VI)-ion imprinted polymer (U(VI)-IIPs) was created using a modified version methodology of Behbahani et al. [37]. In the first step, uranyl acetate was combined with Br-PADAP in a mixture of acetonitrile and dimethyl sulfoxide to create a print ion template. Next, a radical polymerization was carried out using methacrylic acid (MMA) as the functional monomer and ethylene glycol dimethacrylate (EGDMA) as a cross-linker, catalyzed by AIBN [38]. Following the ion-imprinted polymerization, uranium was removed using nitric acid, which left behind cavities that matched the size and shape of the imprinted metal ion.

The properties of a polymer containing ions are heavily influenced by the conditions in which the polymerization reaction occurs, as well as the characteristics of the metal complex involved [39]. Behbahani et al. were able to create a metal complex using Br-PADAP and uranyl ions. Their findings indicated that the electronic structure of a dye molecule could not be significantly affected by the electrostatic interactions of a bound metal ion. However, they discovered that the observed effect was caused by a change in the conjugation system of the Br-PADAP molecule after being complexed with uranyl ions, resulting in a longer conjugation system [40].

The analysis of the scanning electron microscopy (Figure 3) results of the three synthesized polymers indicates that the unprinted polymer has a rough surface without pores, whereas the polymers with imprinted ions have agglomerates of particles of different sizes and shapes, with high porosity. This structure can be attributed to selective binding sites of the uranyl ion [41,42]. The addition of metallic salts to the polymerization medium results in a multiphase separation of the polymer and produces porosity in the matrices [41].

Through Fourier transform infrared spectrum analysis (Figure 4), it is observed that the leached (U(VI)-IIP-lix) and non-leached (U(VI)-IIP) ion-printed polymers exhibit practically the same characteristic bands as the unprinted polymer (NIP), indicating that the complexation and leaching did not affect the main structure of the polymer (NIP). The leached polymer shows an increase in absorption in the band at 1389 cm^−1^, indicating the presence of the 5-Br-PADAP ligand of the complex-U(VI) before and after U(VI) ion removal after leaching [42].

According to the thermogravimetric curves, three thermal events can be observed. The weight loss at low temperatures (<100 °C) is due to the loss of weakly bound water, while the mass loss at 150–250 °C corresponds to the decomposition of residual organic compounds. The third mass loss is attributed to the decomposition of the polymer structure occurring in two steps in the range of 250 to 450 °C, which differs among the synthesized polymers. The addition of the U(VI) complex causes a decrease in the peak and an increase in the shoulder, indicating the thermal stabilization of the polymer structure by complexation (U(VI)-IIP). After leaching (U(VI)-IIP-Leach), a mass loss centered around 270 °C corresponds to the decomposition of residual compounds, and the decomposition of the U-printed polymeric structure occurs in a single event, as a small peak at 420 °C [37]. The total weight loss of the polymers at 700 °C corresponds to 85%, 86%, and 88% for NIP, IIP, and IIP-leach, respectively. The graphs representing these results can be seen in Figure 5.

### 2.4. Optimization of Experimental Conditions for Uranium Extraction and Determination

The preconcentration system containing a 500 mg mini-column of IIP-U(VI) was constructed for the determination of uranium in natural waters using digital imaging. The experimental variables sample pH, eluent concentration, and sampling flow were optimized using a two-level full factorial design [43,44,45,46,47] and a Doehlert matrix [41,43,48]. The experimental domains as coded and actual values of these factors established in the factorial design are shown in Table 2.

The data obtained by factorial design were evaluated and the results found can be seen in Table 3. Of the three factors studied, two were significant as well as their interactions. The pH factor had a positive effect and the concentration of nitric acid had a negative effect.

The curvature was also calculated based on the difference between the average of the responses (absorbance) obtained from the experiments in the full factorial design (0.1946) and the average of the responses from the central point experiments (0.2795). The result (−0.0849) reveals that in addition to statistical significance, there is a region of maximum analytical signal close to the central point experimental conditions [49].

From the analysis of the Pareto chart, new planning was carried out with the help of the Doehlert matrix, to obtain the optimal conditions for the pre-concentration and determination of U(VI). The matrix with the coded and real values together with the responses in the absorbance signal can be seen in Table 4. In Figure 6, the response surface generated from the values in the table below can be seen.

The results obtained generated a model that showed an optimal experimental condition for the pre-concentration of uranium at pH 5.7 and 0.22 mol L^−1^ in the eluent concentration. The red region indicated in Figure 6 graphically represents these values.

To evaluate the behavior of the system in the face of variations around the values of optimal conditions, a robustness test was performed [49]. The same was conducted by applying a complete two-level factorial design, where the experimental domain was varied by ± 20% around the optimal values. Thus, the factors pH and eluent concentration (HNO_3_) ranged from 4.5 to 6.8 and 0.17 to 0.26 mol L^−1^, respectively.

With the Pareto graphic (Figure 7), it can be seen that there were no significant variations at a confidence level of 95%, thus making it possible to say that the proposed method is robust for the experimental domain studied.

### 2.5. Validation Studies

The optimized conditions allowed the determination of uranium in natural water samples with detection and quantification limits of 2.55 and 8.51 μg L^−1^, respectively. These limits were determined using a sample volume of 25 mL and were calculated as recommended by the IUPAC, where LOD = 3δ/α and LOQ = 10δ/α, where δ is the standard deviation of ten analytical blank measurements and α is the slope from the calibration curve obtained by the pre-concentration system [50]. The analytical curve has linearity in the range of 8.51 to 500 μg L^−1^. The pre-concentration factor (PF) found was 8.2. It was calculated by the ratio between the slope obtained by the pre-concentration system and the slope found by the external calibration technique. The precision of the preconcentration system was evaluated using a 50 µg L^−1^ uranium solution by measurements of five replicates (Table 5). The result expressed in relative standard deviation (RSD%) was 3.5%.

The new proposed method may be compared with others already published [51,52,53,54,55,56] (Table 5).

The new proposed method has a similar detection limit to the other batch ones. The ICP-MS is more sensitive, but much bigger and more expensive instruments are required. The spectrophotometric systems are much more sensitive but require instrumentation which only allows it to be used in the lab. The newly proposed method may be easily adapted to be portable and used in field measurements.

### 2.6. Application

The proposed pre-concentration system was used for the determination of uranium in natural waters. Four samples were collected in the city of Caetité (Bahia, Brazil). The results obtained, expressed in confidence intervals, are shown in Table 6. It can be seen that the uranium concentration found ranged from >LQ to 0.075 mg L^−1^. The accuracy of the method was evaluated by addition/recovery tests, which were performed for all four samples analyzed with the addition of 50, 60, and 100 µg L^−1^ of uranium ions. The recoveries found ranged from 91% to 104%.

## 3. Experimental

### 3.1. Reagents and Solutions

All reagents used in the experiments are analytical grade, and ultra-pure water was obtained using a Milli-Q Plus system (Bedford, MA, USA). Ethylene glycol dimethacrylate (EGDMA), 2-(5-bromo-2-pyridylazo)-5-diethylaminophenol (5-Br-PADAP), methacrylic acid (AMA), 2,2′-azobisisobutyronitrile (AIBN), and uranyl acetate (UO_2_(CH_3_COO)_2_·2H_2_O) were supplied by (Merck, Darmstadt, Germany). Uranium solutions were prepared daily by diluting a 1000 mg L^−1^ stock solution (Merck, Darmstadt, Germany). Nitric acid solutions were prepared by diluting 67% nitric acid (HNO_3_) (MERCK, Darmstadt, Germany). Dihydrogen potassium orthophosphate/sodium hydroxide and sodium acetate/acetic acid buffer solutions were prepared with a concentration of 0.1 mol L^−1^, both (NEON, Suzano, Brazil). The glasses and containers used were washed and kept in a nitric acid solution (10% *v*/*v*) for 24 h.

Water samples from rivers and lakes were collected in the city of Caetité, Bahia, filtered through a cellulose acetate membrane with a diameter of 47.0 mm and porosity of 0.45 μm, acidified, and stored in a refrigerator at 4 °C until analysis.

### 3.2. RGB Data Acquisition and Evaluation

The determination of U(VI) was performed using a closed system consisting of a closed white wooden box (to avoid reflections), with the following dimensions: 35 cm × 40 cm × 25 cm (height × width × depth). The box also had a light control system in the internal compartment to ensure reproducibility during image capture. Attached to this box was a Lifecam Cinema 720 p 5 MB webcam, model H5d-00013 from Microsoft, as can be seen in Figure 8.

For all experiments, digital images were stored in JPG format and a square region of interest (ROI) was defined at a fixed position using the ImageJ computer program treatment. This program allows the acquisition of RGB data for all pixels in the ROI. This information is organized into a color histogram, and the average value of each color channel, red (R), green (G), and blue (B), is calculated. The definition of the analytical signals based on the color value in accordance with Bee’s law was defined as: (Equation (1))
A = log (P_0_/P)(1)
where P is the R, G, and B value (mean or mode) of the standard solution or sample and P_0_ is the R, G, and B value for the analytical blank.

In the batch pre-concentration system, an MS Tecnopon LDP 305-4 peristaltic pump equipped with silicone tubes of various diameters was used to propel the solutions.

### 3.3. Synthesis of Polymers with Printed Ions (IIP-U(VI))

The polymer synthesis with imprinted ions was based on the work of Dhruv K. Singh [40]. It was carried out using 0.25 mmol of uranyl acetate (UO_2_(CH_3_COO)_2_·2H_2_O) and 0.25 mmol of 5-Br-PADAP which were solubilized in 10 mL of a 1:1 mixture of dimethylsulfoxide/acetonitrile with continuous stirring for one hour. After this period, the resulting solution was mixed with 4.0 mmol of AMA, 20 mmol of EGDMA, and 0.100 g of the radical initiator AIBN. With the addition of such reagents, the polymerization was continued by heating in an oil bath at 60 °C for a period of 24 h. The material at the end of the synthesis reaction was dried, crushed, sieved, and leached in consecutive steps using ethanol and 2.0 mol L^−1^ hydrochloric acid. The unprinted polymer (NIP) was prepared in a similar way but without the U(VI) ions. Finally, the materials obtained were characterized by SEM, FTIR, and TG. Figure 9 represents the proposed synthesis route of the IIP-U(VI).

### 3.4. Characterization of the Ion-Imprinted Polymer (IIP)

The morphology of the polymers was analyzed by scanning electron microscopy (SEM), using a JEOL microscope, Carry Scopy JSM-6510LV model (JEOL, Ltd., Tokyo, Japan), with an acceleration voltage of 20 kV. The samples were mounted on aluminum sample holders and coated with gold using evaporative plating equipment, specifically a DESK V model (Denton Vacuum, NJ, USA). Thermogravimetric analysis (TGA) was conducted using equipment from TA Instruments, Q50 model (TA Instruments, New Castle, DE, USA) with a heating rate of 10 °C/min in a nitrogen atmosphere, over a temperature range of 27 to 800 °C. Fourier transform infrared (FTIR) analysis was performed using a Nicolet iS 10 instrument from Thermo Scientific (Thermo Scientific, Waltham, MA, USA) with a spectral range from 4000 to 400 cm^−1^, a resolution of 4 cm^−1^, and 32 scans.

### 3.5. Preconcentration System and Determination of U(VI)

The procedure for pre-concentration of U(VI) ions using IIP-U(VI) was carried out in batches and consisted of two steps. In step 1 (pre-concentration), 25 mL of the sample was passed through a mini-column containing 0.1 g of the sorbent. In the second step, 2.0 mL of the eluent (nitric acid) was percolated through the mini-column to carry all the ions of interest that were previously absorbed in step 1.

To the eluted volume (2.0 mL) was added 1.0 mL of the chromogenic reagent (Arsenazo III) with a definite concentration of 0.05% (*w*/*v*). This final solution was taken to the detection system (Figure 1). A schematic of the pre-concentration and elution system is illustrated in Figure 10.

### 3.6. Multivariate Optimization Strategy

The optimization of the online system was performed using a full two-level factorial design and a Doehlert matrix. All experiments were performed in a random order using uranium solution with final concentrations of 2.0 and 1.0 mg L^−1^, respectively. The variables pH, eluent concentration (nitric acid), and sampling flow were chosen for the optimization and absorption as a response. Triplicates of the central point were used to calculate the experimental error.

### 3.7. Sample Preparation

Samples of water from lakes in the region of the city of Caetité (Bahia, Brazil) were collected and stored in polyethylene bottles. The samples were filtered using a filter with a cellulose acetate membrane of 47.0 mm in diameter and 0.45 μm in porosity. With subsequent filtration, the samples were acidified and stored in a refrigerator at 4 °C until analysis. In Table 7 are found the geographic coordinates.

This region of the municipality of Caetité (Bahia, Brazil), which is located 645 km from the capital of the state of Salvador, Ba, has an approximate population of 51,000 inhabitants, according to IBGE 2019 [35]. It was chosen for the collection of lake water samples. This region is subject of many studies aimed at the analysis of possible uranium contamination since it is naturally located in a uranium mine, with reserves of 100,000 tons of the ore being explored by the state-owned Indústrias Nucleares do Brasil SA (INB) since 1999, which is capable of producing around 400 tons of uranium/year [36].

## 4. Conclusions

The proposed method proved to be efficient in the determination of U(VI) in lake water samples, presenting excellent precision and accuracy, as well as adequate detection and quantification limits for the concentrations frequently found in these samples.

The chemometric tools used to study the optimal conditions of the factors studied proved to be efficient in the face of all variations in the experimental domains.

The methodology developed for the determination of U(VI) ions through digital images presented itself as an excellent alternative for the quantification of these ions, mainly because of the ease of assembling the system, in addition to data processing (image decomposition into RGB signals) being possible through free software such as Chemostat. Finally, the synthesized polymer allowed the extraction and pre-concentration of U(VI) ions from highly complex matrices such as lake waters.

## Figures and Tables

**Figure 1 molecules-28-04065-f001:**
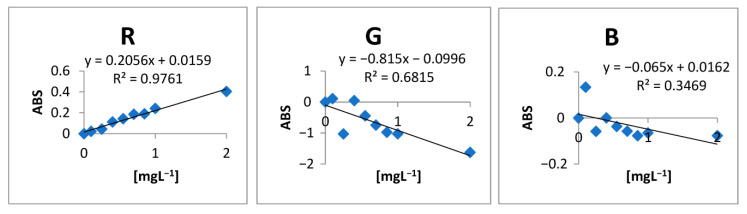
Analytical curves of the different RGB channels. R = red; B = blue; G = green.

**Figure 2 molecules-28-04065-f002:**
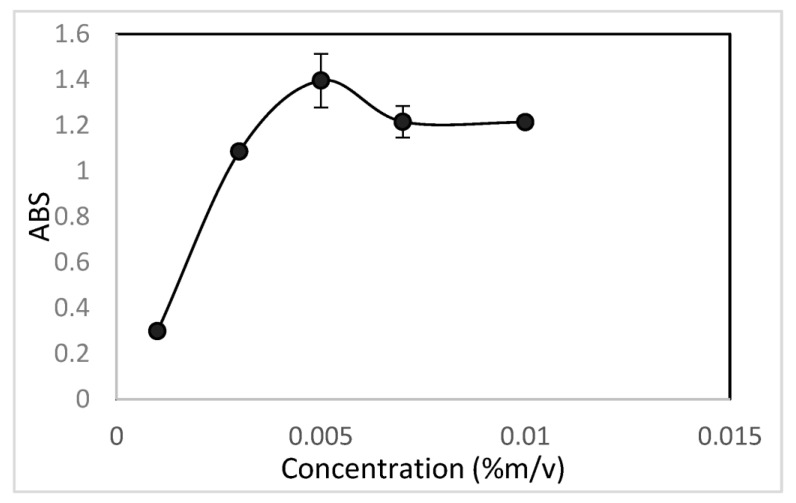
Study of the effect of concentration variation on the U(VI)–arsenazo III complex.

**Figure 3 molecules-28-04065-f003:**
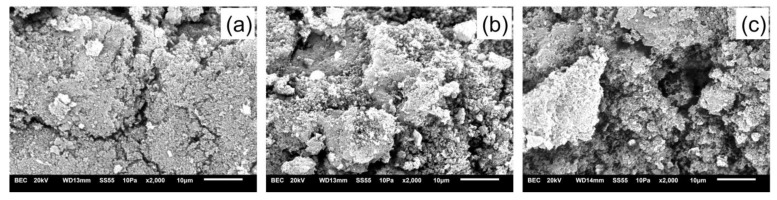
Scanning electronic microscopy (SEM) of synthesized polymers. (**a**) Unprinted polymer (NIP); (**b**) non-leached polymer (U-IIP); (**c**) leached polymer (U-IIP-lix).

**Figure 4 molecules-28-04065-f004:**
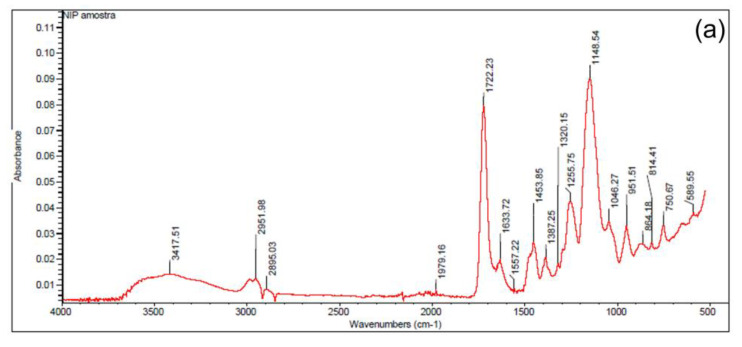
FTIR of synthesized polymers. (**a**) Unprinted polymer (NIP); (**b**) non-leached polymer (U-IIP); (**c**) leached polymer (U-IIP-lix).

**Figure 5 molecules-28-04065-f005:**
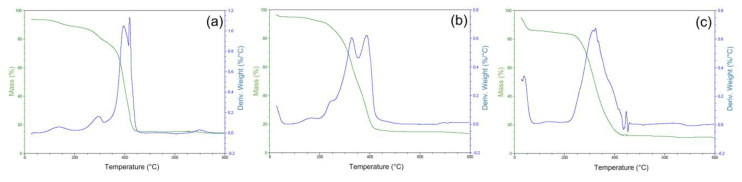
TG graph of synthesized polymers. (**a**) Unprinted polymer (NIP); (**b**) non-leached polymer (U-IIP); (**c**) leached polymer (U-IIP-lix).

**Figure 6 molecules-28-04065-f006:**
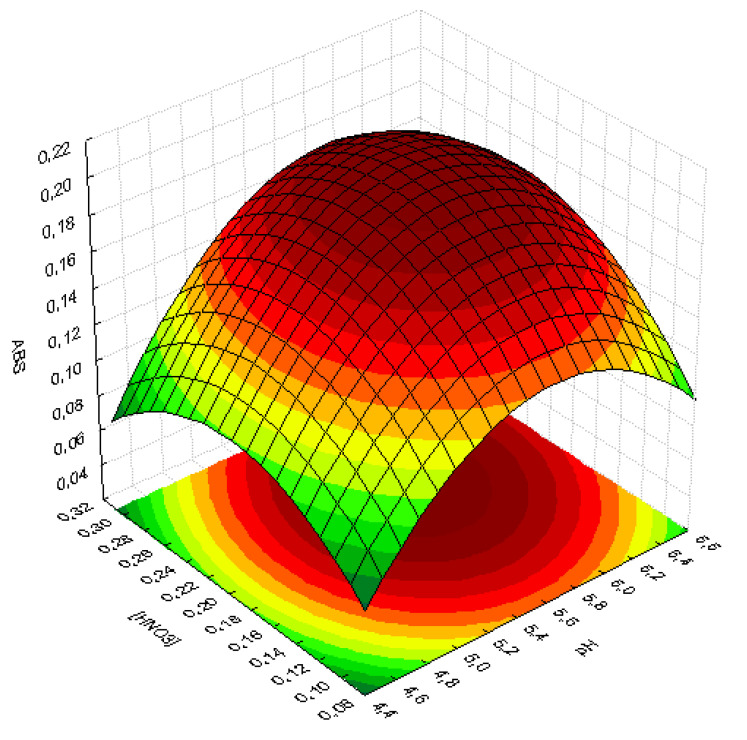
Response surfaces obtained by the Doehlert design for concentrated nitric acid and pH.

**Figure 7 molecules-28-04065-f007:**
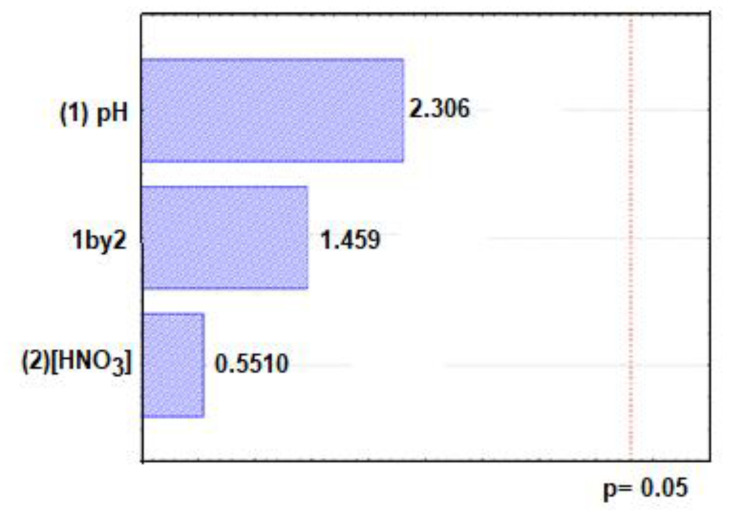
Pareto graphic of the robustness test.

**Figure 8 molecules-28-04065-f008:**
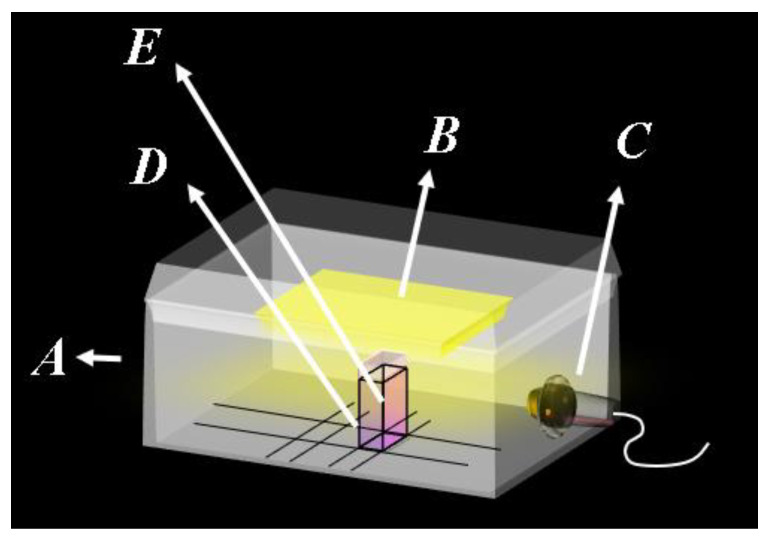
Model of the box used for the acquisition of digital images of the proposed methodology. A: Box; B: lamp 6 W SMD LED white light 6500 k; C: cam; D: sample holder; E: bucket.

**Figure 9 molecules-28-04065-f009:**
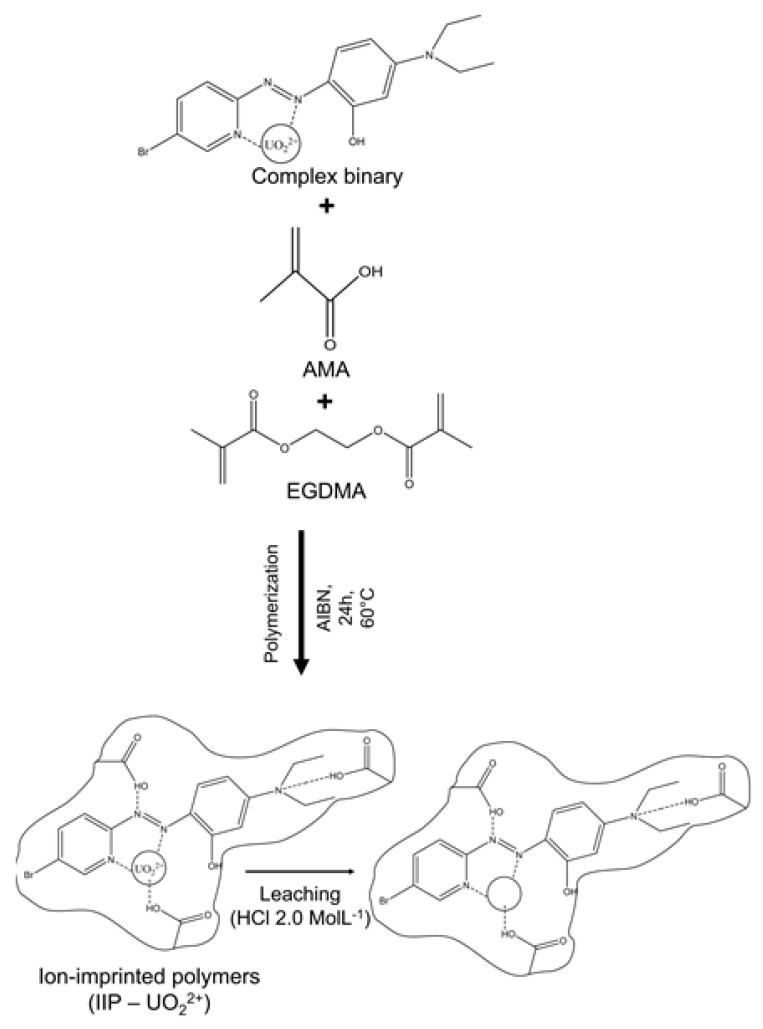
Schematic representation of the proposed synthesis route of the IIP-UO^2+^.

**Figure 10 molecules-28-04065-f010:**
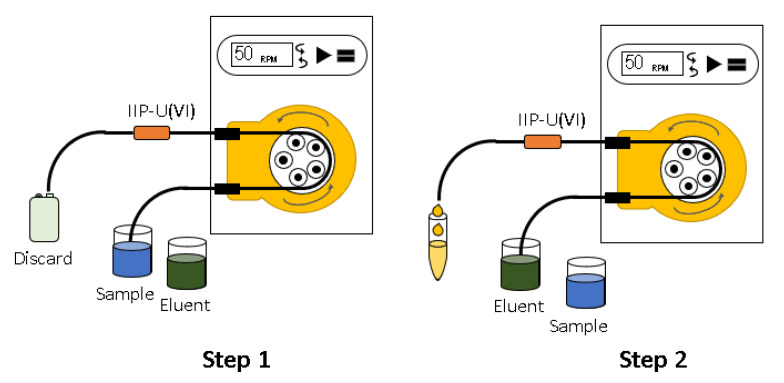
Batch system for preconcentration using the IIP-U(VI) (**Step 1**), and elution with nitric acid (**Step 2**).

**Table 1 molecules-28-04065-t001:** RGB values and their respective absorbance values.

Conc. (mg L^−1^)	R	G	B	Abs R	Abs G	Abs G
0	157.85	0.58	55.80	0	0	0
0.1	149.86	0.45	40.90	0.02	0.11	0.13
0.25	142.61	6.16	63.76	0.04	−1.03	−0.06
0.40	121.74	0.52	55.70	0.11	0.05	0
0.55	113.40	1.61	60.69	0.14	−0.44	−0.04
0.70	102.73	3.21	63.63	0.19	−0.74	−0.06
0.85	101.81	5.46	66.58	0.19	−0.97	−0.08
1.0	90.47	6.13	64.71	0.24	−1.02	−0.06
2.0	62.18	24.41	66.55	0.40	−1.62	−0.08

**Table 2 molecules-28-04065-t002:** Matrix of the complete two-level factorial design used for the preliminary evaluation of the factors involved in the extraction and determination of U(VI).

Experiment	pH	Sampling Flow	[HNO_3_]	Absorbance
1	− (2.5)	− (2.0)	− (0.1)	0.024
2	− (2.5)	− (2.0)	+ (0.5)	0.009
3	− (2.5)	+ (6.0)	− (0.1)	0.014
4	− (2.5)	+ (6.0)	+ (0.5)	0.009
5	+ (5.5)	− (2.0)	− (0.1)	0.428
6	+ (5.5)	− (2.0)	+ (0.5)	0.335
7	+ (5.5)	+ (6.0)	− (0.1)	0.418
8	+ (5.5)	+ (6.0)	+ (0.5)	0.316
9	0 (4.0)	0 (4.0)	0 (0.3)	0.275
10	0 (4.0)	0 (4.0)	0 (0.3)	0.275
11	0 (4.0)	0 (4.0)	0 (0.3)	0.287

**Table 3 molecules-28-04065-t003:** Effects of factors and their interactions on the bismuth extraction.

Factor and Interaction	Effects
pH	0.360 ± 0.0048
[HNO_3_]	−0.053 ± 0.0048
pHx[HNO_3_]	−0.0437 ± 0.0048

**Table 4 molecules-28-04065-t004:** Doehlert matrix used to optimize the U(VI) preconcentration method.

Experiment	pH	[HNO_3_]	Absorbance
1	0 (5.5)	0 (0.2)	0.203
2	0 (5.5)	0 (0.2)	0.190
3	0 (5.5)	0 (0.2)	0.192
4	1 (6.5)	0 (0.2)	0.182
5	0.5 (6.0)	0.866 (0.3)	0.165
6	−1 (4.5)	0 (0.2)	0.110
7	−0.5 (5.0)	−0.866 (0.1)	0.148
8	0.5 (6.0)	−0.866 (0.1)	0.139
9	−0.5 (5.0)	0.866 (0.3)	0.155

**Table 5 molecules-28-04065-t005:** Detection limit of the proposed method compared with others already published papers.

Detection System	Method	LOD (μg L^−1)^	Ref.
Radiometric	Batch	18.0	51
	Batch	2.5	52
	Batch	0.4	53
ICP-MS	FIA	0.7	54
	FIA	0.02	55
Spectrophotometric	MSFIA-MPFS	0.013	56
	LOV-MSFIA	0.01	20
Digital imaging	Batch	2.55	Our method

**Table 6 molecules-28-04065-t006:** Addition and recovery test.

Samples	Addition (µg L^−1^)	Concentration (µg L^−1^)	Recovery (%)
LG01	0	75 ± 5.10	-
	100	180 ± 2.44	104
LG03	0	>LQ	-
	50	45.6 ± 6.20	91
P04	0	20.3 ± 4.21	-
	50	75 ± 4.0	109
R04	0	35 ± 4.2	-
	60	35 ± 4.2	92

**Table 7 molecules-28-04065-t007:** Geographical coordinates of sample collection points.

Sample	Coordinates
LG01	Lat. −13.837463°—Long. −42.311125°
LG03	Lat. −14.300701°—Long. −42.578394°
P04	Lat. −14.100147°—Long. −42.435525°
R04	Lat. −14.165014°—Long. −42.441480°

Note: Lat. = Latitude, Long. = Longitude.

## Data Availability

Data is contained within the article.

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
