# Peer review of "Synthesis and Application of a New Polymer with Imprinted Ions for the Preconcentration of Uranium in Natural Water Samples and Determination by Digital Imaging"

_molecules, 2023, doi:10.3390/molecules28104065_

Round 1
Reviewer 1 Report
This manuscript describes the Synthesis and Application of a New Polymer with Imprinted 3 Ions for the Preconcentration of Uranium in Natural Water Samples and Determination by Digital Imaging.
The work should be in the interest of the reader of the journal. The results are interesting and potentially useful. However, there are some errors (both English language and technical) that the authors need to consider before they can be published.
1: What are the novelty, or unique ideas behind this research as compared to previous research/other reported work? Why it is worth knowing?
2: The typographic errors should be corrected throughout the whole manuscript, for example reference [i, ii] cited in wrong format.
3: Uranium salt company name is not mentioned. Washing or leaching of uranium from polymer with HCL and ethanol, the author did not mention how they have disposed the solution containing leached uranium?
4: Please the interaction mechanism between the active site and the template ion should be incorporated into the graphical abstract.
5: SME images need more discussion about morphological changes.
6: EDX analysis may be included to support the data.
7: The physicochemical stability of IIP must be investigated.
8: The matrix effect must be checked for real water samples in order to confirm its selectivity.
9: Imprinting factor, adsorption capacity of imprinted polymer (mgg-1) must be added.
10: The below references could be added in the introduction part to strengthen the introduction section of ion imprinted polymer.
https://doi.org/10.1016/j.microc.2019.02.037
https://doi.org/10.1007/s10965-020-02196-0
Comments as above.
Author Response
Please see pdf attachment.

Reviewer 2 Report
In this manuscript Molecules-2355879,the authors tried to synthesize new polymer with with imprinted ions (IIP) and used for the determination of uranium. The polymer was prepared by using 2-(5-Bromo-2-pyridylazo)-5-diethyla-16 minophenol (Br-PADAP) for complex formation, ethylene glycol dimethacrylate (EGDMA) as a crosslinking reagent, methacrylic acid (AMA) as functional monomer, and 18 2,2’-azobisisobutyronitrile was used as a radical initiator. I would like to recommend it publish in Molecules after the following revisions.
1. Please redraw high resolution Figures in all manuscript. Most of the pictures are not clear, please increase the higher resolution of the pictures (such as Figure 3, 6, 7, 8).
2. The LOD for the determination of uranium should compare with other literature.
3. Most of the references need to be updated, and the latest research literature is selected.
4.Line 192 the formation of the U(VI) - arsenazo III complex is 0.05 % (m/v), the concentration is 0.005 % in Figure 5,which one is correct?
Author Response
Please see pdf attachment.
